# StreamNet: Memory-Efficient Streaming Tiny Deep Learning Inference on the Microcontroller

**Hong-Sheng Zheng, Chen-Fong Hsu, Yu-Yuan Liu, Tsung Tai Yeh**
Department of Computer Science
National Yang-Ming Chiao Tung University
Hsinchu, Taiwan
{hszheng.cs08, fonghsu.cs08, yyliu.cs11, ttyeh14}@nycu.edu.tw

## Abstract

With the emerging Tiny Machine Learning (TinyML) inference applications, there is a growing interest when deploying TinyML models on the low-power Microcontroller Unit (MCU). However, deploying TinyML models on MCUs reveals several challenges due to the MCU's resource constraints, such as small flash memory, tight SRAM memory budget, and slow CPU performance. Unlike typical layer-wise inference, patch-based inference reduces the peak usage of SRAM memory on MCUs by saving small patches rather than the entire tensor in the SRAM memory. However, the processing of patch-based inference tremendously increases the amount of MACs against the layer-wise method. Thus, this notoriously computational overhead makes patch-based inference undesirable on MCUs. This work designs StreamNet that employs the stream buffer to eliminate the redundant computation of patch-based inference. StreamNet uses 1D and 2D streaming processing and provides an parameter selection algorithm that automatically improve the performance of patch-based inference with minimal requirements on the MCU's SRAM memory space. In 10 TinyML models, StreamNet-2D achieves a geometric mean of 7.3X speedup and saves 81% of MACs over the state-of-the-art patch-based inference.

## 1 Introduction

With the emerging Tiny Machine Learning (TinyML) applications, challenges to successfully deploy TinyML models on the resource-constrained microcontroller (MCU) are becoming critical. Unlike the high-performance CPU, the MCU is basically composed of a small flash memory (several MBs), an SRAM memory (hundreds of KBs), and a slow CPU (hundreds of MHz). Since the SRAM memory has better performance on reading and writing data than the flash memory, most TinyML system frameworks often place the input/output tensors on the SRAM and store weights and filters on the flash memory. In addition, the size of memory on MCUs is always small because of their constrained budgets. However, recent TinyML models tend to use large input images and intermediate tensors to raise their training accuracy (1; 2). Such TinyML models expose grand challenges when deploying these models on MCUs with the stringent memory constraint.

Tensor memory planner of TinyML system frameworks determines the usage of the SRAM memory. An MCU is often responsible for Machine Learning (ML) inference and often uses a single batch to facilitate the response time of ML inference tasks. Thus, the static tensor memory planner schedules intermediate tensors to meet the SRAM memory constraint. Conventionally, the compiler/interpreter employs layer-wise scheduling, pushes data of a layer in the SRAM memory, and replaces them after finishing its operator (3; 4). However, such layer-wise inference will overuse the SRAM memory space when TinyML models have large intermediate tensors. To overcome this shortcoming, patch-

based inference (2) divides the entire tensors into multiple overlapping patches. Then, patch-based inference places a patch in the SRAM memory space rather than the entire tensor to repair the out-of-memory problem. However, the redundant computation of patch-based inference occurs when proceeding with a large number of overlapping patches and tremendously increases the execution time of TinyML models. Furthermore, this situation becomes worse in particular when significantly shrinking the usage of SRAM memory space. Hence, patch-based inference is becoming undesirable with such notoriously high performance overhead on MCUs.

This work designs *StreamNet* that eliminates the significant amount of redundant computations while reaching minimal usage of the SRAM memory space for TinyML models on MCUs. StreamNet creates the stream buffer to reuse tensor data in the patch-based inference. In addition, StreamNet designs 1D and 2D stream processing to improve the performance of patch-based inference with different SRAM memory requirements. To facilitate the performance of StreamNet further, StreamNet also removes the unnecessary computation on the padding data. At last, StreamNet's parameter selection framework automatically constructs a new patch-based inference that mixes 1D and 2D stream processing to meet the memory constraint while resulting in the best performance for TinyML models.

The main contributions of this paper are shown below:

1. StreamNet achieves a geometric mean of 3.4X (1D) to 7.3X (2D) speedup over the existing patch-based inference without increasing memory usage significantly.

2. Developing output padding bypass kernels to enhance the performance of patch-based inference.

3. The StreamNet parameter selection framework automatically selects an optimal configuration for StreamNet-2D operations under the SRAM memory constraint of an MCU.

## 2 Understanding Performance Bottleneck of Memory Allocation on TinyML

**Memory-efficient patch-based inference**: Patch-based inference aims to reduce the peak SRAM memory usage of TinyML models on MCUs. This patch-based approach divides a large input feature map into multiple small patches. Then, the MCU's SRAM buffer reduces its space usage by only storing a small patch each time instead of the entire input feature map. For instance, Figure 1(a) demonstrates the peak SRAM memory usage of the proxyless-w0.3 TinyML model (5) and other TinyML models have the similar trend. Traditionally, ML system frameworks such as TFLM (3) place input and output tensors of a layer or an operator in the SRAM buffer. After finishing the work of a layer, TFLM (3) removes the obsolete input data of the completed layer and allocates another space to put the output data of the next layer. However, given an MCU and its size of SRAM memory is 128 KB, Figure 1 demonstrates that the memory usage of multiple layers in the proxyless-w0.3 TinyML model (5) is larger than the size of the SRAM memory on an MCU. As a result, the layer-wise memory allocation raises significant challenges when deploying TinyML models on MCUs. Unlike the layer-wise inference, Figure 1 presents patch-based inference that reduces the total SRAM memory usage of TinyML models and enables models to be run on MCUs with small memory size.

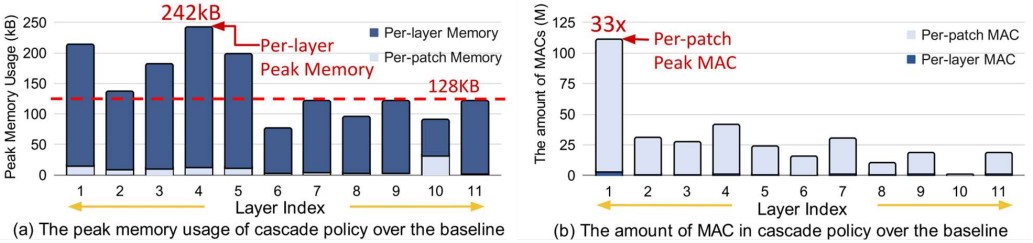

(a) The peak memory usage of cascade policy over the baseline    (b) The amount of MAC in cascade policy over the baseline

Figure 1: Comparing peak memory usage and the amount of MACs on patch-based and layer-wise inference

**Challenges and opportunities**: Patch-based inference creates a large amount of redundant computation that occurs among its adjacent patches. To reduce the peak SRAM memory usage on MCUs,

patch-based inference creates many small patches that overlap receptive fields. These small patches tremendously raise the amount of computation on a TinyML model. The inefficient patch-based inference is becoming undesirable on MCUs even if it saves the peak SRAM memory usage of a TinyML model significantly. Figure 1(b) demonstrates that patch-based inference takes 33X more MACs than the layer-wise one at the first layer. Furthermore, the amount of MACs in patch-based inference also rapidly grows with the reduction of peak SRAM memory usage. Hence, our work stores repeated results of patch-based inference in the stream buffer and skips these computations to eliminate the performance bottleneck of patch-based inference.

## 3 StreamNet: Efficient Patch-based Inference for TinyML Models

This section details the design and implementation of StreamNet that removes the redundant computation of patch-based inference with the minimal usage of the SRAM memory space on MCUs.

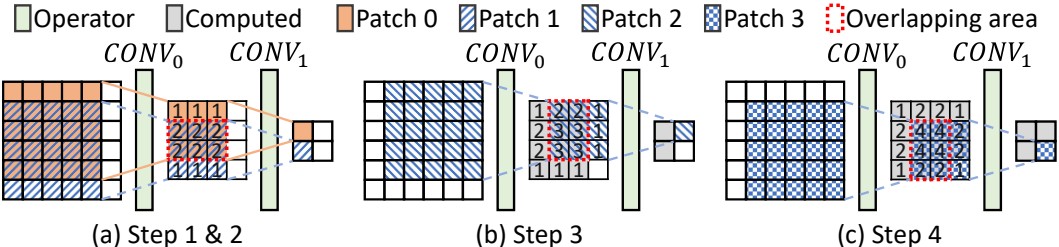

Figure 2: Patch-based inference example

### 3.1 Inefficient Patch-based Inference

To accelerate patch-based inference, this work proposes StreamNet that removes most redundant computations of patch-based inference by putting its repetitive results in the stream buffer. Figure 2 demonstrates operations of patch-based inference. In Figure 2, there are two operators where each of them has its own input and output tensor and the computed block means the completed output element. Conventionally, in the convolution processing, the $6 \times 6$ input with the $3 \times 3$ filter will yield the $4 \times 4$ output. This calculation takes 144 ($16 \times 9$) MACs where the number 16 is the size of the output and the number 9 comes from the $3 \times 3$ filter. Unlike conventional convolution, in Figure 2, patch-based inference divides a $6 \times 6$ input into four $5 \times 5$ patches. Each $5 \times 5$ patch computes the convolution with the $3 \times 3$ filter to finally yield a $4 \times 4$ output. As illustrated in Figure 2(a), Patch 0 and Patch 1 generate a $3 \times 3$ output patch, respectively. Each of them takes $9 \times 9$ MACs where the first number 9 is the size of output and the second number 9 is associated with the size of the filter. Thus, patch-based inference uses 4 patches to create a $4 \times 4$ output and takes 324 MACs ($9 \times 9 \times 4$). The amount of MACs of patch-based inference is 2.25X (324/144) more than the layer-wise inference because patch-based inference contains an enormous amount of repetitive computations. For instance, in Figure 2, each value in the overlapping area represents the repetition count of the output tensor element. This redundant calculation increases the execution time of a TinyML model significantly. In addition, patch-based inference tends to use the fine-grained patch to lower the usage of the SRAM memory. However, the amount of redundant computation increases as the patch size becomes small.

### 3.2 StreamNet 1D Streaming Processing

Figure 3(a) illustrates the steps of 1D streaming processing on StreamNet inference. In Figure 3(a), two patches proceed one after the other as the data stream during the computation of the convolution. In Figure 3, the computed block represents the completed element of the tensor, and the computing block indicates the elements of the tensor in the execution. At first, in Figure 3(a), StreamNet-1D performs the CONV0 operator on the $5 \times 5$ patch and yields a $3 \times 3$ output tile (❶). Second, StreamNet-1D stores the overlapping area of this $3 \times 3$ output tile in the stream buffer (❷). Then, StreamNet-1D uses this $3 \times 3$ tile to generate the $1 \times 1$ output data after finishing the CONV1 operator (❸). Unlike patch-based inference, StreamNet uses the results preserved in the stream buffer during the computation of Patch 1. Thus, StreamNet-1D only fetches a $3 \times 5$ tile from Patch 1 to yield a

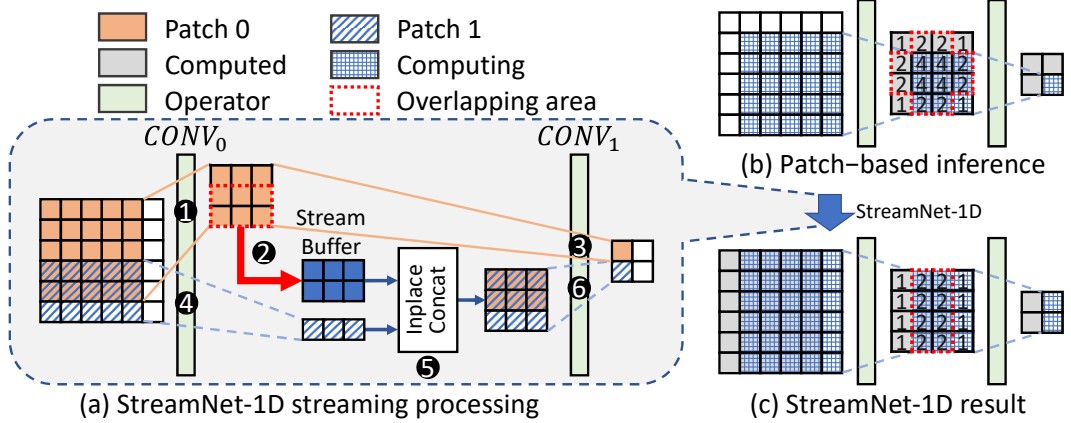

Figure 3: The operation of StreamNet-1D

$3 \times 1$ output data (❹). Then, StreamNet-1D combines this $3 \times 1$ output with the data of the stream buffer to work out a $3 \times 3$ output tile as illustrated in Figure 3(a) (❺). At last, the $3 \times 3$ output of Patch 1 will yield a $1 \times 1$ output after finishing the CONV1 operator (❻). In the computation of Patch 0 and Patch 1, StreamNet-1D takes 108 MACs ($12 \times 9$) where the number 12 is the sum of the calculated output size in (❶) ($3 \times 3$) and (❹) ($3 \times 1$) and the number 9 is associated with the size of the $3 \times 3$ filter. Since StreamNet-1D also needs to use four patches to create a $4 \times 4$ output in total, the total amount of MACs in 1D streaming processing of StreamNet-1D is 216 ($12 \times 9 \times 2$). Unlike patch-based inference in Figure 3(b), StreamNet-1D in Figure 3(c) decreases the repetition counts in each element of the output tensor. Hence, StreamNet-1D reduces roughly 60% of MAC overhead over patch-based inference in Figure 3, because the overhead of StreamNet-1D and patch-based inference is 8 (24 - 16) and 20 (36 - 16) where 16 ($4 \times 4$) is the number of output elements of the layer-wise inference.

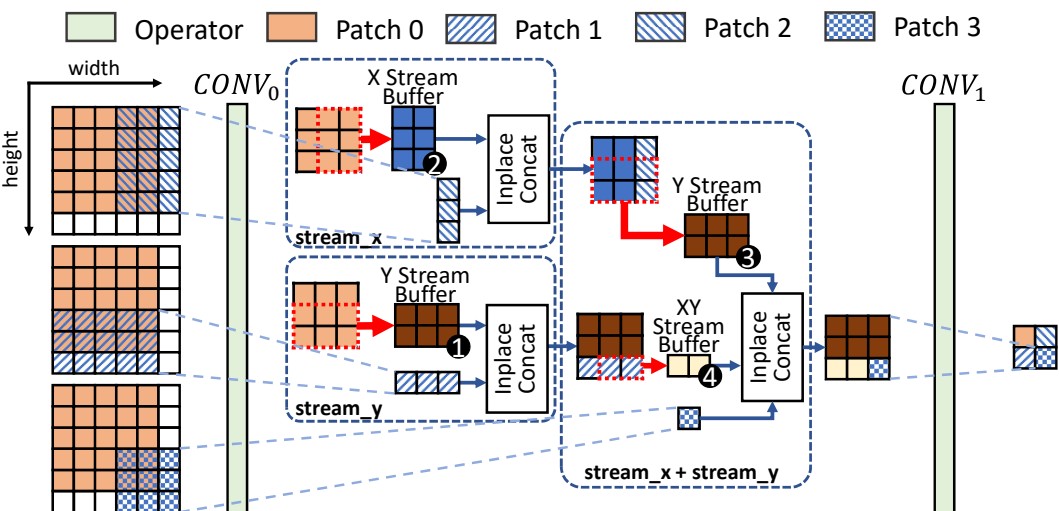

Figure 4: The StreamNet-2D operations

## 3.3 StreamNet 2D Streaming Processing

StreamNet uses the stream buffer to preserve temporary results that will be used multiple times across patches. After finishing an operator, StreamNet will place a number of output tensors in the stream buffer. Since StreamNet-1D does not completely remove the redundant computation overhead, StreamNet designs StreamNet-2D to eliminate all redundant MACs in patch-based inference. StreamNet-2D is composed of stream_x and stream_y processing. The execution order of

stream_y patches is firstly moving downward. For instance, patch-based inference in Figure 4 moves its execution vertically for Patch 1, then back to Patch 2, and finally moves downward for Patch 3. Unlike stream_y, the execution order of stream_x patch firstly moves toward the horizontal direction. At first, in Figure 4, the $5 \times 5$ Patch 0 computes CONV0 with the $3 \times 3$ filter and yields a $3 \times 3$ output. Then, StreamNet-2D stores a $3 \times 2$ tile and $2 \times 3$ tile of the $3 \times 3$ output in the Y and X stream buffer (❶)(❷), respectively. Second, StreamNet-2D tackles Patch 1 and fetches a $5 \times 3$ tile in Patch 1 to yield a $3 \times 1$ output. After finishing Patch 1, StreamNet stores a $2 \times 1$ tile of its output tensor to the XY stream buffer (❹) and cleans the Y stream buffer. Third, StreamNet-2D handles Patch 2 and reuses results in the X stream buffer. After completing Patch 2, StreamNet-2D stores a $3 \times 2$ tile of Patch 2's output in the Y stream buffer (❸)). Finally, StreamNet-2D reuses the data within the Y and XY stream buffer to proceed with Patch 3. StreamNet-2D takes 144 MACs ($(9+3+3+1) \times 9$) where each number of the first bracket represents the amount of calculation in each patch and the second number 9 means the size of the $3 \times 3$ filter. Hence, StreamNet-2D completely eliminates the total amount of the redundant computation shown in patch-based inference. However, StreamNet-2D has a larger stream buffer to store the temporary results of each patch over StreamNet-1D. As illustrated in Figure 4, the stream buffer of StreamNet-2D is 14 where the $2 \times 3$ tile is from stream_x, the $3 \times 2$ tile is from stream_y, and the $2 \times 1$ tile is from the XY stream buffer. A large stream buffer increases the requirement of the SRAM memory space. To balance the consumption of the MACs and the SRAM memory space, StreamNet presents an parameter selection framework that mixes StreamNet-1D and StreamNet-2D. As a result, StreamNet can satisfy the constraint of the SRAM memory size on MCUs while achieving efficient execution on TinyML models.

---

**Algorithm 1:** The StreamNet Parameter Selection Algorithm

---

**Input:** $model$ and $constraint$
**Output:** selected StreamNet parameter under constraint
1   $patch\_params = \{(split\_idx, n\_patch) \mid split\_idx \in ValidSplitIndexes(Model),$
                              $n\_patch \in OutShapeFactors(Layers[split\_idx])\}$
2   $candidates = [\,]$
3   **for** *param in patch_params* **do**
4     $performance = (mem, MACs) = ProfileParameter\_PatchBased(param)$
5     **if** *mem meets constraint* **then**
6       $candidates.append(param, performance)$
7   Filter obtained $records$ and only keep the ones that have $performance$ on the Pareto frontier
8   **for** *candidate in candidates* **do**
9     **for** *stream_x_level in [0, 1, ..., record.split_idx]* **do**
10       $param = (candidate.param, max\_stream\_y\_level, stream\_x\_level)$
11       $performance = (mem, MACs) = ProfileParameter\_StreamNet(param)$
12       **if** *mem meets constraint* **then**
13         $new\_candidates.append(param, performance)$
14   **if** *there is only one param that has the smallest $MACs$* **then**
15     **return** $param$ as the best parameter.
16   **else**
17     **return** the $param$ that has the largest output patch size as the best parameter.

---

### 3.4 StreamNet Parameter Selection Framework

Searching for the best parameter that satisfies memory constraints while minimizing the latency of a TinyML model manually is exceedingly tedious and challenging. To address this problem, StreamNet designs an parameter selection framework that determines the best parameter candidate to meet the constraint of the SRAM memory across different MCUs. Hence, StreamNet parameter selection framework aims to automatically find the parameter composition of StreamNet and mixes StreamNet-1D and StreamNet-2D to fit the limited amount of SRAM memory on the MCU. To minimize the runtime overhead, StreamNet refers to the MAC count to choose the best candidate of StreamNet parameters through its compiler pass.

Algorithm 1 demonstrates StreamNet parameter selection framework that infers parameters of StreamNet to fit an MCU at the compile time. The patch parameter in line 1 of Algorithm 1 includes the number of the patch (n_patch) and the number of layers that proceed with patch-based

computation (split_idx). The OutShapeFactor function returns the valid number of the patch in patch-based inference. Next, StreamNet calculates the amount of MACs with different combinations of these two parameters and picks out candidates that meet the constraint of the SRAM memory space of an MCU from line 2 - 7 of the Algorithm 1. Then, StreamNet examines different values of stream_x_level and stream_y_level to determine the number of stream-wise layers while having a minimum amount of MACs in a model and returns this combination from line 8 - 15 of Algorithm 1. Otherwise, StreamNet tends to choose the one with the smallest patch count when both candidates have the same amount of MACs, since a large output patch has bigger computation granularity, which usually yields better latency.

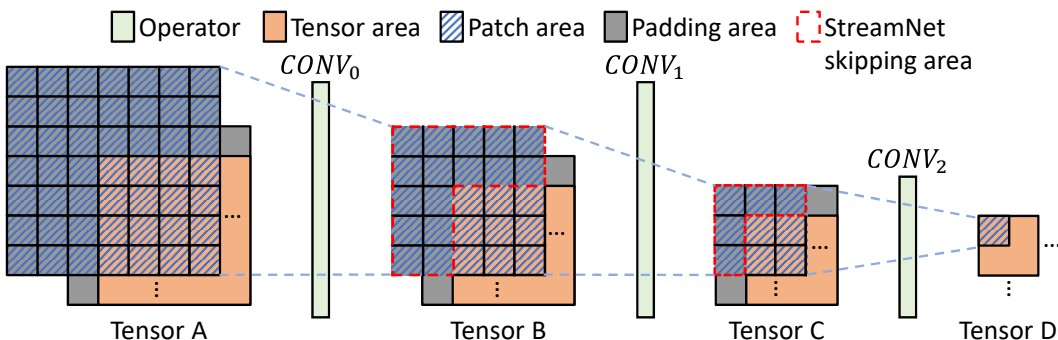

Figure 5: Output padding data bypassing on StreamNet

## 3.5 Output Padding Bypassing

The padding data of each tensor contributes the performance overhead in the patch-based inference. To further the performance of patch-based inference, StreamNet intelligently skips the computation of the padding data. Figure 5 presents a patch-based inference neural network that consists of 3 operators and 4 tensors. Padding is always added to a tensor to generate an output with the specified size. Parameters of conventional neural networks such as the size of the stride and the filter result in different amounts of padding data in each tensor. Unlike common neural networks, in Figure 5, the patch size of the current tensor is obtained from the configuration of the next tensor. For instance, patch-based inference refers to the patch size of Tensor C to figure out the patch size of Tensor B. The padding data in Tensor B appends the one in Tensor C. Thus, the amount of the padding data in patch-based inference can be more than the one in the layer-wise inference. Since the value of padding data is always assigned to be zero, the padding data won't change the expected results. In Figure 5, patch-based inference proceeds with a $7 \times 7$ patch in Tensor A including padding data to generate an output $5 \times 5$ patch. Next, this $5 \times 5$ patch in Tensor B is used to create a $3 \times 3$ patch in Tensor C. Hence, each operator computes its padding data that contributes additional MACs during the execution of TinyML models. StreamNet removes the performance overhead shown in the padding data by skipping the computation of each output padding data. In Figure 5, patch-based inference reads the data of a $5 \times 5$ patch in Tensor B to generate a $3 \times 3$ patch in Tensor B. Unlike patch-based inference, StreamNet uses the size of the padding data known after finishing the previous layer to distinguish the padding data from the real one. Next, StreamNet only considers the real data in the calculation of an operator. Furthermore, StreamNet only writes the zero value into output padding data directly without consuming any MACs. StreamNet changes the implemented kernel of patch-based inference to skip the computation of the output padding data during the execution of each operator.

## 4 StreamNet System Architecture

The system architecture of StreamNet contains the frontend and backend processing. In the frontend system of StreamNet, the input of StreamNet is the TensorFlow Lite model (.tflite file). Then, StreamNet uses the parameters of the TinyML model to create layers of patch-based inference (2). Second, the memory scheduler allocates SRAM memory space for tensors on the static arena memory space and then StreamNet performs the 1D and 2D stream processing with its stream buffers. In StreamNet's backend system, StreamNet translates tensors and operators of the patch-based inference

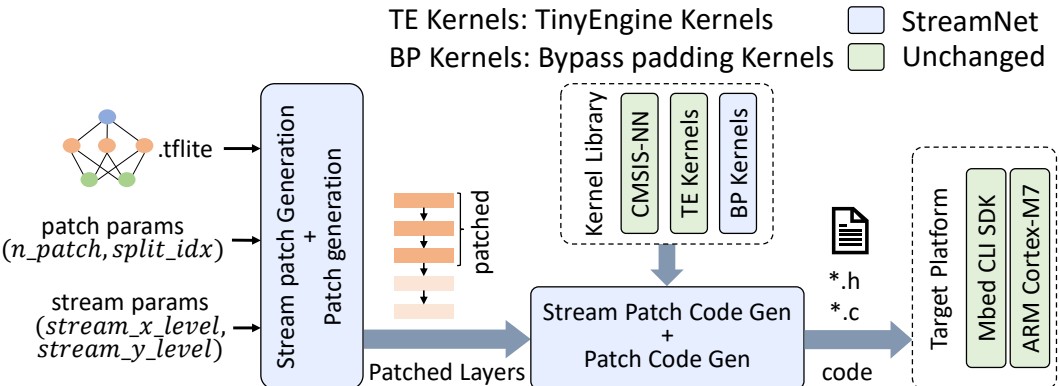

Figure 6: The system architecture of StreamNet

into C source code and links kernel libraries such as ARM CMSIS-NN (6), TinyEngine (1; 2), and our bypass padding kernels. Finally, StreamNet deploys TinyML models on the MCU by using Mbed CLI SDK (7).

## 5 Experiments

### 5.1 Experiment Environment Setup

**System Implementation**: We compare the performance and memory consumption of StreamNet to MCUNetV2 (2), which performs patch-based inference on TinyML models. Since existing TinyEngine kernels (1) used by MCUNetV2 (2) do not support $5 \times 5$ and $7 \times 7$ filters, we fix this issue to enable our evaluation to work on the depthwise convolution function with more filter combinations. In addition, we also insert our output padding bypassing implementation within MCUNetV2 (2) as MCUNetV2+. StreamNet modifies MCUNetV2's memory scheduler and code generator to identify streamable area, allocate stream buffer, generate codes with different streaming processing, and insert the stream buffer operation between operator kernel invocation in StreamNet.

**Benchmark**: We evaluate 10 TinyML models from MCUNetV2 model zoo (5)such as mbv2-w0.35(MB2), proxyless-w0.3(PL), mcunet-vww0(MV0), mcunet-vww1(MV1), mcunet-vww2(MV2), mcunet-in0(MI0), mcunet-in1(MI1), mcunet-in2(MI2), mcunet-in3(MI3), mcunet-in4(MI4). All models use int8 quantized mode. These TinyML models have different requirements on the SRAM memory space and their execution time also differs.

**On-device Experiment**: We deploy TinyML models through ARM Mbed CLI (7) on the ARM Cortex-M7 CPU. The MCU used in our evaluation is STM32F767ZI (8) that includes an ARM Cortex-M7 CPU at 216 MHz, a 512KB SRAM, and a 2MB Flash

### 5.2 Performance Analysis when Consuming Nearly Equal Memory

Table 1 compares the performance of StreamNet and MCUNetv2 when both of them use nearly the same SRAM memory space. Table 1 presents the data of StreamNet-2D. Then, we scan through the value of n_patch and split_idx parameters to search for the one where their SRAM memory usage is closest to StreamNet-2D. In Table 1, StreamNet achieves a geometric mean of 5.11X speedup over MCUNetv2. The SRAM memory consumption varies with the change of n_patch and split_idx parameters on MCUNetv2. However, the amount of MACs increases rapidly with the reduction of SRAM memory usage on MCUNetv2. As a result, StreamNet has better performance than MCUNetv2 when consuming nearly equal SRAM memory.

### 5.3 The Variation of MAC Counts

Table 2 compares the amount of MACs used in each TinyML model. In Table 2, the value of the n_patch and split_idx determines the number of patches and the number of patch-based layers used in a model, respectively. StreamNet scans through all combinations of the n_patch and split_idx and

Table 1: Performance results of MCUNetv2 and StreamNet-2D when consuming nearly equal memory. Param (n_patch, split_idx), Memory (KB), MAC (millions), Latency (ms)

| Model | StreamNet-2D | | | | MCUNetV2 | | | | Speedup | |
| | Param | Mem | MAC | Latency | Param | Mem | MAC | Latency | MAC | Latency |
|---|---|---|---|---|---|---|---|---|---|---|
| MB2 | (18,13) | 66 | 24 | 417 | (18,12) | 63 | 77 | 1,513 | 3.29 | 3.63 |
| PL | (22,13) | 95 | 38 | 676 | (11,21) | 89 | 549 | 9,519 | 14.31 | 14.08 |
| MI0 | (12,06) | 30 | 6 | 99 | (04,14) | 29 | 19 | 274.441 | 2.99 | 2.77 |
| MI1 | (12,11) | 47 | 13 | 188 | (03,12) | 48 | 24 | 366 | 1.84 | 1.95 |
| MI2 | (20,21) | 169 | 67 | 1,168 | (05,25) | 169 | 463 | 7,055 | 6.87 | 6.04 |
| MI3 | (22,21) | 208 | 82 | 1,444 | (11,32) | 215 | 2,777 | 43,812 | 33.89 | 30.35 |
| MI4 | (20,17) | 236 | 126 | 1,762 | (04,21) | 214 | 366 | 4,613 | 2.91 | 2.62 |
| MV0 | (08,13) | 29 | 6 | 101 | (08,11) | 30 | 17 | 296 | 2.83 | 2.93 |
| MV1 | (20,14) | 44 | 12 | 225 | (05,17) | 45 | 47 | 810 | 4.04 | 3.61 |
| MV2 | (18,21) | 143 | 56 | 961 | (06,29) | 144 | 518 | 8,311 | 9.27 | 8.65 |
| GMEAN | | | | | | | | | 5.33 | 5.11 |

Table 2: The amount of MACs (Millions) used in StreamNet and MCUNetv2

| Model | n_patch | split_idx | MNv2 | MNv2+ | SN-1D | SN-2D | Speedup over MNv2 | | |
| | | | | | | | MNv2+ | SN-1D | SN-2D |
|---|---|---|---|---|---|---|---|---|---|
| MB2 | 18 | 13 | 77.4 | 73.1 | 36.9 | 23.5 | 1.06 | 2.10 | 3.29 |
| PL | 22 | 13 | 349.6 | 326.5 | 88.6 | 38.4 | 1.07 | 3.95 | 9.11 |
| MI0 | 12 | 06 | 15.5 | 14.4 | 8.5 | 6.4 | 1.08 | 1.82 | 2.43 |
| MI1 | 12 | 11 | 27.0 | 25.4 | 16.6 | 12.8 | 1.06 | 1.62 | 2.11 |
| MI2 | 20 | 21 | 998.8 | 920.2 | 189.2 | 67.4 | 1.09 | 5.28 | 14.82 |
| MI3 | 22 | 21 | 1,784.7 | 1,650.7 | 271.7 | 81.9 | 1.08 | 6.57 | 21.78 |
| MI4 | 20 | 17 | 1,942.1 | 1,787.8 | 353.8 | 126.0 | 1.09 | 5.49 | 15.41 |
| MV0 | 08 | 13 | 23.1 | 20.0 | 9.7 | 6.0 | 1.15 | 2.39 | 3.87 |
| MV1 | 20 | 14 | 96.1 | 89.1 | 24.2 | 11.6 | 1.08 | 3.97 | 8.28 |
| MV2 | 18 | 21 | 826.8 | 754.3 | 153.3 | 55.9 | 1.10 | 5.39 | 14.79 |
| GMEAN | | | | | | | 1.08 | 3.45 | 7.18 |

finds the one that can result in the minimal usage of SRAM memory space in patch-based inference. Thus, results in Table 2 show the amount of MACs used in the TinyML model with the lowest peak SRAM memory usage through all possible combinations of n_patch and split_idx. The amount of the MACs in patch-based inference increases with the reduction of the usage on the SRAM memory. Therefore, MCUNetv2 (MNv2) obtains a significant amount of the MACs over the other alternatives in Table 2. In addition, MCUNetv2+ (MNv2+) reduces roughly 8% of MACs on average over the baseline MCUNetv2. The output padding bypassing of MCUNetv2+ removes overhead on the padding data and achieves such a reduction on the amount of MACs. Moreover, StreamNet-1D (SN-1D) performs 1D stream_y processing and saves the redundant computation through its stream buffer. Thus, StreamNet-1D reduces 65% of MACs on average over the baseline. To completely remove the redundant computation on patch-based inference (MCUNetv2), StreamNet-2D (SN-2D) performs 2D streaming processing to increase more data reuse than StreamNet-1D. Hence, StreamNet-2D reduces 83% of MACs on average over MCUNetv2 across 10 TinyML models. StreamNet-2D saves 93% and 95% of MACs on MI2 and MI3, since these models consist of many patch-based layers with the large split_idx value and the amount of MACs on the redundant computation is large.

## 5.4 Reveal the Latency of StreamNet

This evaluation aims to figure out if StreamNet can shorten the execution time of TinyML models by reducing the amount of their MACs. Table 3 compares the latency of each TinyML model across different implementations on MCUNetv2(MNv2) and StreamNet. The n_patch and split_idx in Table 3 and Table 2 use the same value. Table 3 shows the execution time when TinyML models use minimum SRAM memory space through patch-based inference. MCUNetv2+ (MNv2+), StreamNet-1D (SN-1D), and StreamNet-2D (SN-2D) achieves a geometric mean of 1.1X, 3.4X, 7.3X speedup

Table 3: The latency (ms) of TinyML models on StreamNet and MCUNetv2

| Model | MNv2 | MNv2+ | SN-1D | SN-2D | Speedup over MNv2 | | |
|---|---|---|---|---|---|---|---|
| | | | | | MNv2+ | SN-1D | SN-2D |
| MB2 | 1,514 | 1,391 | 687 | 417 | 1.09 | 2.20 | 3.63 |
| PL | 6,296 | 5,795 | 1,610 | 676 | 1.09 | 3.91 | 9.31 |
| MI0 | 238 | 219 | 131 | 99 | 1.09 | 1.81 | 2.40 |
| MI1 | 449 | 413 | 258 | 188 | 1.09 | 1.74 | 2.39 |
| MI2 | 16,545 | 15,055 | 3,281 | 1,168 | 1.10 | 5.04 | 14.17 |
| MI3 | 29,700 | 27,331 | 4,858 | 1,444 | 1.09 | 6.11 | 20.57 |
| MI4 | 26,816 | 24,747 | 5,153 | 1,762 | 1.08 | 5.20 | 15.22 |
| MV0 | 415 | 355 | 169 | 101 | 1.17 | 2.45 | 4.10 |
| MV1 | 1,817 | 1,667 | 473 | 225 | 1.09 | 3.84 | 8.09 |
| MV2 | 14,106 | 13,040 | 2,778 | 961 | 1.08 | 5.08 | 14.68 |
| GMEAN | | | | | 1.10 | 3.40 | 7.28 |

over the baseline MCUNetv2. The speedup of Table 3 presents the consistent results over the amount of MACs in Table 2. Hence, this result provides StreamNet the evidence to exploit the MAC count obtained at compile time to infer the final latency in StreamNet auto-tuning framework.

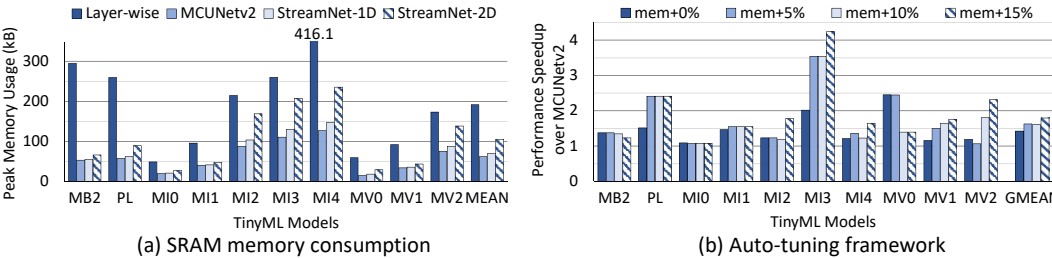

(a) SRAM memory consumption

(b) Auto-tuning framework

Figure 7: Results of SRAM Memory consumption and StreamNet's auto-tuning framework

## 5.5 Analysis of Peak SRAM Memory Usage

Figure 7(a) presents the peak SRAM memory usage of TinyML models on different implementations of MCUNetv2 (2) and StreamNet. MCUNetv2 (2), StreamNet-1D, and StreamNet-2D save 66%, 62%, and 47% SRAM memory usage on average over the layer-wise inference baseline, respectively. Unlike patch-based inference, StreamNet-1D reuses the data within the stream buffer that occupies the SRAM memory space. Thus, StreamNet-1D consumes additional 4% SRAM memory space on average to store the stream buffer over the MCUNetv2 (2). To increase the ratio of the data reuse, StreamNet-2D requires about 22% extra SRAM memory space on average for the stream buffer to cover more data reuse than StreamNet-1D. StreamNet provides a trade-off between the performance and the peak usage of the SRAM memory. Unlike patch-based inference, StreamNet improves the performance of TinyML models without sacrificing too much SRAM memory space.

## 5.6 Results of StreamNet Auto-tuning Framework

This evaluation aims to validate if StreamNet auto-tuning framework can create a case that satisfies its memory constraint while achieving the best performance. Figure 7(b) presents the performance speedup results when adding 0% to 15% memory space on each TinyML model based on the minimal SRAM memory usage of StreamNet-1D. In addition, we also design an auto-tuning framework for MCUNetV2 (2) that scans through its n_patch and split_idx parameters to obtain a case with the best performance while meeting the memory constraint. Unlike MCUNetV2 (2) auto-tuning framework, StreamNet searches for the combination of StreamNet-1D and StreamNet-2D network to fit the memory constraint while achieving the best performance. As illustrated in Figure 7(b), StreamNet achieves a geometric mean of 1.63X, 1.61X, 1.8X speedup over MCUNetV2 (2) when the memory constraint is the base memory space plus 5%, 10%, and 15%, respectively. In Figure 7(b), MI0

achieves only a speedup of 1.07X to 1.09X over MCUNetV2 (2), where the value of the split_idx and n_patches is small, leading to minor recomputation in MCUNetV2 (2). Unlike MI0, MI3 uses larger split_idx and n_patches values to meet the memory constraints. The large value of split_idx and n_patches allows StreamNet to eliminate the significant amount of redundant computation.

## 6   Related Work

**Tiny Machine Learning (TinyML)**   TinyML has gained significant attention in recent years. Techniques such as model compression (9; 10), pruning (11; 12; 13; 14; 15) and quantization (11; 16; 17; 18; 19; 20; 21; 22), are employed to reduce model size and complexity while maintaining accuracy. Existing frameworks like TFLM (3) and microTVM (4) optimize the execution time of TinyML models on resource-constrained devices by utilizing CMSIS-NN (6) and CMix-NN (23) libraries for kernel implementation. MCUNet (1) further optimizes kernel implementation through vectorization, loop reordering, loop unrolling, and data layout transformation, resulting in more efficient performance.

**Memory-Efficient Inference**   The peak usage of SRAM memory is a critical factor in deploying TinyML models on microcontrollers due to the tight SRAM constraints in MCUs. However, existing works (3; 24; 25) employ per-layer inference, which restricts model deployment on severely memory-constrained MCUs. Several approaches have been proposed to address this issue in different ways. Some methods rearrange operators (26; 27) or use tensor memory planners (28) to save memory usage. Nonetheless, a single intermediate tensor may still be too large for the MCUs. As an alternative, models can be rewritten to reduce memory requirements (29; 30). However, this requires network modifications and may affect accuracy, reducing adaptability. MCUNetV2 (2) and TVM's cascade scheduling (31) address this problem by fusing multiple layers and introducing redundant computation. Prior work (32; 33; 34; 35; 4) also tackles similar problems as MCUNetv2 (2) in the ML inference. However, too many fused layers increase the amount of redundant computation.

**Remove Recomputations**   To address recomputations introduced by the aforementioned approach, the main idea is to cache the computed result and reuse it when necessary. Halide (35) introduces sliding window optimization to remove recomputation, but it lacks flexibility in adjusting temporary buffers to reduce memory requirements across different dimensions. Prior work such as Fused-layer CNN accelerators (36) and streaming methods (37; 38; 39; 40; 41) offer solutions to reduce execution time in the data streaming manner, but none of them provides a method to mix 1D and 2D data streaming. Hence, our proposed StreamNet builds upon and extends existing streaming approaches by introducing a hybrid method that mixes different dimensional streaming to form 1D and 2D streaming techniques, skips the redundant computation shown in patch-based inference (2), and offers an auto-tuning framework to select StreamNet configurations.

## 7   Conclusion

Deploying TinyML models exposes challenges on MCUs with the stringent memory and power constraints. Unlike layer-wise inference, patch-based inference significantly decreases the usage of the SRAM memory and helps the deployment of TinyML models on MCUs. However, the redundant computation of patch-based inference enormously raises the latency of Machine Learning inference. Thus, this work proposes StreamNet that aims to eliminate such a performance bottleneck of patch-based inference. StreamNet designs 1D and 2D streaming processing to significantly remove the performance overhead of patch-based inference. Thus, StreamNet-2D achieves a geometric-mean speedup of 7.3X over patch-based inference across 10 TinyML models. Furthermore, StreamNet also skips the unnecessary computation for the padding data. Finally, StreamNet provides an auto-tuning framework to automatically search for the network that can satisfy the memory constraint of MCUs while achieving better performance over the existing patch-based inference.

## 8   Acknowledgement

The authors gratefully acknowledge the support from the National Science and Technology Council in Taiwan under grant number 111-2221-E-A49-131-MY3

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
