# Supplementary Material
# StreamNet: Memory-Efficient Streaming Tiny Deep Learning Inference on the Microcontroller

## Contents

37th Conference on Neural Information Processing Systems (NeurIPS 2023).

# A System Architecture

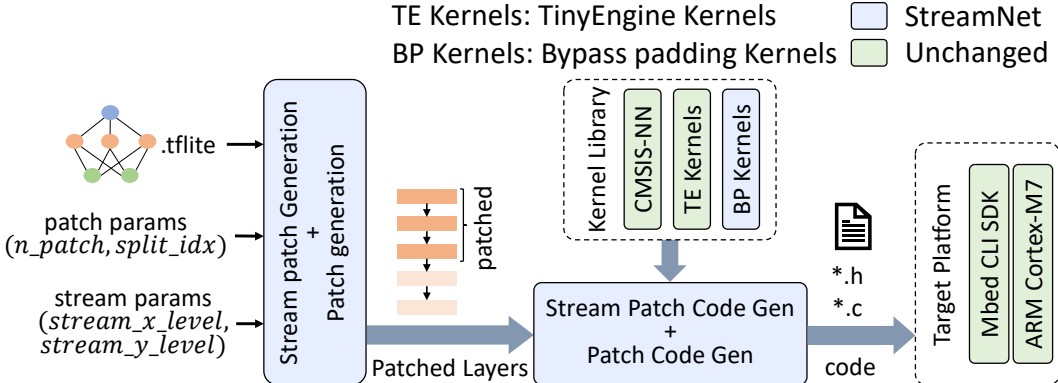

Figure 1: The system architecture of StreamNet

TensorFlow Lite for Microcontrollers (TFLM) (1) tailors for the TinyML applications and adopts the interpreter-based approach to make cross-platform interoperability in the embedded system possible. However, TFLM's interpreter increases the performance overhead of the TinyML applications on MCUs. Unlike TFLM, StreamNet and MCUNetv2 replace the interpreter with a code generator. StreamNet is built on top of MCUNetv2 (2; 3) and adds the feature of the 1D and 2D stream processing (4; 5; 6; 7; 8; 9; 10). The code generator of StreamNet produces kernel implementations with fixed parameters at the compile time. Then, codes generated by StreamNet can be well-optimized by the backend C/C++ target compiler through constant folding and loop unrolling. As a result, StreamNet decreases the runtime overhead by using its code generator to tackle tensors of TinyML models (11).

The system architecture of StreamNet contains the frontend and backend processing. In the frontend system of StreamNet, the input of StreamNet is the TensorFlow Lite model (.tflie file). Then, StreamNet uses the parameters of the TinyML model to create layers of patch-based inference (3). Second, the memory scheduler allocates SRAM memory space for tensors on the static arena memory space and then StreamNet performs the 1D and 2D stream processing with its stream buffers. In StreamNet's backend system, StreamNet translates tensors and operators of the patch-based inference into C source code and links kernel libraries such as ARM CMSIS-NN (12), TinyEngine (2; 3), and our bypass padding kernels. Finally, StreamNet deploys TinyML models on the MCU by using Mbed CLI SDK (13).

# B Performance Analysis when Consuming Nearly Equal Memory

Table 1 compares the performance of StreamNet and MCUNetv2 when both of them use nearly the same SRAM memory space. Table 1 presents the data of StreamNet-2D. Then, we scan through the value of n_patch and split_idx parameters to search for the one where their SRAM memory usage is closest to StreamNet-2D. In Table 1, StreamNet achieves a geometric mean of 5.11X speedup over MCUNetv2. The SRAM memory consumption varies with the change of n_patch and split_idx parameters on MCUNetv2. However, the amount of MACs increases rapidly with the reduction of SRAM memory usage on MCUNetv2. As a result, StreamNet has better performance than MCUNetv2 when consuming nearly equal SRAM memory.

# C MAC/Latency Correlation Analysis

To reduce the runtime overhead, StreamNet uses the amount of multiply-and-accumulate (MACs) of TinyML models collected at the compile time to guide its auto-tuning framework. This evaluation aims to validate the correctness of StreamNet that uses the offline data rather than the real latency result of TinyML models on the MCU. Figure 2 presents the magnitude of the overhead on the layer-wise inference compared to our StreamNet. Each dot in Figure 2 indicates all possible results of streaming parameters across 10 TinyML models. We calculate the Pearson product-moment

Table 1: Performance results of MCUNetv2 and StreamNet-2D when consuming nearly equal memory. Param (n_patch, split_idx), Memory (KB), MAC (millions), Latency (ms)

| Model | StreamNet-2D | | | | MCUNetV2 | | | | Speedup | |
|---|---|---|---|---|---|---|---|---|---|---|
| | Param | Mem | MAC | Latency | Param | Mem | MAC | Latency | MAC | Latency |
| MB2 | (18,13) | 66 | 24 | 417 | (18,12) | 63 | 77 | 1,513 | 3.29 | 3.63 |
| PL | (22,13) | 95 | 38 | 676 | (11,21) | 89 | 549 | 9,519 | 14.31 | 14.08 |
| MI0 | (12,06) | 30 | 6 | 99 | (04,14) | 29 | 19 | 274.441 | 2.99 | 2.77 |
| MI1 | (12,11) | 47 | 13 | 188 | (03,12) | 48 | 24 | 366 | 1.84 | 1.95 |
| MI2 | (20,21) | 169 | 67 | 1,168 | (05,25) | 169 | 463 | 7,055 | 6.87 | 6.04 |
| MI3 | (22,21) | 208 | 82 | 1,444 | (11,32) | 215 | 2,777 | 43,812 | 33.89 | 30.35 |
| MI4 | (20,17) | 236 | 126 | 1,762 | (04,21) | 214 | 366 | 4,613 | 2.91 | 2.62 |
| MV0 | (08,13) | 29 | 6 | 101 | (08,11) | 30 | 17 | 296 | 2.83 | 2.93 |
| MV1 | (20,14) | 44 | 12 | 225 | (05,17) | 45 | 47 | 810 | 4.04 | 3.61 |
| MV2 | (18,21) | 143 | 56 | 961 | (06,29) | 144 | 518 | 8,311 | 9.27 | 8.65 |

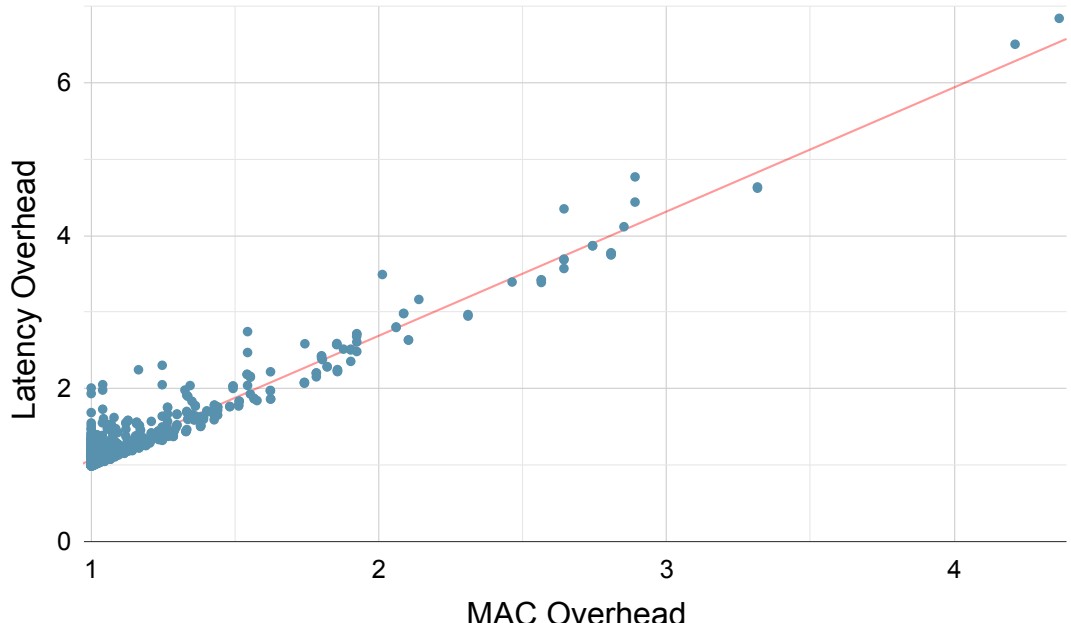

Figure 2: The correlation of MAC over latency on TinyML Models

correlation coefficient to figure out the relationship between the value of MACs and the final execution time of StreamNet. Figure 2 demonstrates that the Pearson correlation coefficient is 0.97 and the Pearson $R^2$ correlation is 0.94. These results show the strong correlation between MAC and latency of TinyML models. Hence, the amount of MACs in the TinyML model is an appropriate candidate that can be used to infer the final latency of TinyML models on our StreamNet.

## D   Latency Analysis of Patch-wise Layers

Patch-wise layers contribute a large percentage of execution time in the TinyML model. The patch-based inference leverages the split_idx to determine the number of patch-wise layers in a TinyML model. Thus, the number of patch-wise layers varies across different TinyML models. This experiment only counts the results of patch-wise layers and helps to figure out the overhead removal of StreamNet on patch-wise layers. In Table 2, StreamNet-2D and StreamNet-1D achieve a geometric mean speedup of 15.78X and 4.45X over the baseline MCUNetv2, respectively. These results are much better than the one shown in the main paper, because this evaluation only considers the patch-wise layers. In addition, results of the execution time on TinyML models are also revealed on the Table 3. In Table 3, StreamNet-1D and StreamNet-2D achieve a geometric mean speedup of

Table 2: The amount of MACs (millions) on patch-wise layers

| model | param | MNv2 | MNv2+ | SN-1D | SN-2D | Speedup over MNv2 | | |
|---|---|---|---|---|---|---|---|---|
| | | | | | | MNv2+ | SN-1D | SN-2D |
| MB2 | (18,13) | 62.75 | 58.67 | 22.47 | 9.15 | 1.07 | 2.79 | 6.86 |
| PL | (22,13) | 326.89 | 303.81 | 65.93 | 15.68 | 1.08 | 4.96 | 20.85 |
| MI0 | (12,06) | 10.80 | 9.70 | 3.83 | 1.69 | 1.11 | 2.82 | 6.40 |
| MI1 | (12,11) | 18.44 | 16.81 | 8.08 | 4.26 | 1.10 | 2.28 | 4.33 |
| MI2 | (20,21) | 966.90 | 888.26 | 157.28 | 35.48 | 1.09 | 6.15 | 27.26 |
| MI3 | (22,21) | 1,743.20 | 1,609.17 | 230.12 | 40.38 | 1.08 | 7.58 | 43.16 |
| MI4 | (20,17) | 1,868.38 | 1,714.09 | 280.01 | 52.28 | 1.09 | 6.67 | 35.73 |
| MV0 | (08,13) | 19.94 | 16.85 | 6.51 | 2.79 | 1.18 | 3.06 | 7.15 |
| MV1 | (20,14) | 88.26 | 81.29 | 16.41 | 3.81 | 1.09 | 5.38 | 23.18 |
| MV2 | (18,21) | 793.83 | 721.38 | 120.32 | 22.93 | 1.10 | 6.60 | 34.62 |
| GMEAN | | | | | | 1.10 | 4.45 | 15.78 |

Table 3: The latency (ms) of patch-wise layers

| model | param | MNv2 | MNv2+ | SN-1D | SN-2D | Speedup over MNv2 | | |
|---|---|---|---|---|---|---|---|---|
| | | | | | | MNv2+ | SN-1D | SN-2D |
| MB2 | (18,13) | 1.34 | 1,215.89 | 512.49 | 242.32 | 1.10 | 2.61 | 5.52 |
| PL | (22,13) | 6,002.88 | 5,516.51 | 1,322.64 | 399.39 | 1.09 | 4.54 | 15.03 |
| MI0 | (12,06) | 181.59 | 161.34 | 75.07 | 43.02 | 1.13 | 2.42 | 4.22 |
| MI1 | (12,11) | 357.97 | 321.37 | 166.86 | 96.89 | 1.11 | 2.15 | 3.69 |
| MI2 | (20,21) | 16,216.19 | 14,721.54 | 2,951.38 | 839.81 | 1.10 | 5.49 | 19.31 |
| MI3 | (22,21) | 29,302.43 | 26,939.08 | 4,440.88 | 1,018.34 | 1.09 | 6.60 | 28.77 |
| MI4 | (20,17) | 26,149.01 | 24,038.13 | 4,470.71 | 1,087.92 | 1.09 | 5.85 | 24.04 |
| MV0 | (08,13) | 378.41 | 316.05 | 132.24 | 64.08 | 1.20 | 2.86 | 5.91 |
| MV1 | (20,14) | 1,726.77 | 1,579.44 | 387.94 | 136.64 | 1.09 | 4.45 | 12.64 |
| MV2 | (18,21) | 13,750.00 | 12,655.84 | 2,431.68 | 613.15 | 1.09 | 5.65 | 22.43 |
| GMEAN | | | | | | 1.11 | 3.96 | 11.12 |

3.96X and 11.12X over the baseline, respectively. As a result, StreamNet indeed accelerates TinyML models by removing redundant computations of overlapping patches in the patch-based inference on MCUs.