# OpenReview forum: "StreamNet: Memory-Efficient Streaming Tiny Deep Learning Inference on the Microcontroller"
_NeurIPS.cc/2023/Conference — NeurIPS 2023 poster_

### Official Review · Reviewer_TjRD · 2023-06-15

**Soundness:** 3 good
**Presentation:** 3 good
**Contribution:** 3 good
**Rating:** 6
**Confidence:** 3

**Summary:**

This paper presents methods for speeding up patch-based inference on microcontrollers. The method creates a buffer to selectively save intermediate values that are traditionally discarded in patch-based computation. This allows StreamNet to balance latency and memory consumption. There are 1d and 2d variants of this optimization.

The paper additionally proposes a method for skipping the computation of padding data and a framework for auto-tuning the hyperparameters of StreamNet. The framework searches for patch hyperparameters that meet certain memory constraints while minimizing latency.

**Strengths:**

- Pure runtime optimization targeted at the most important MCU metrics (latency and memory)
- State-of-the-art performance vs. a strong baseline (MCUNetv2)
- Enables a new dimension for latency-memory tradeoffs
- Includes an algorithm for automatically searching the newly created search space.


**Weaknesses:**

- Given the tradeoff between latency and memory consumption that StreamNet unlocks, it would be easier to interpret results as points on a latency-memory Pareto curve rather than cross-referencing tables and charts. The table in the appendix that gave the speedup at nearly equal memory was the most informative in understanding the benefits, but it was buried.

- Only compares against MCUNetv2 and not the other patch-based optimization mentioned in the related work.

- Relies heavily on existing work (MCUNetv2 and TinyEngine)

- The contributions in the intro should be reworded to be consistent

**Questions:**

- Is this method purely a runtime/compiler optimization, or are there any model architecture implications (e.g. on accuracy)?

- How might one redesign MCU class models to maximize the benefits of StreamNet? If StreamNet more easily optimizes certain layer configurations, how does that impact exiting Pareto curves?

- Will StreamNet be open-sourced?

**Limitations:**

How long does it take to run the auto-tuning framework? How does it compare to black-box search methods?

---

> ### Author Rebuttal · Authors · 2023-08-09
>
> Thank you all for the valuable comments!  In our revised version, we  addressed all of the reviewer comments. The details of our reply and the changes are presented in the following.
>
> D.1 Is this method purely a runtime/compiler optimization, or are there any model architecture implications (e.g. on accuracy)?
>
> Yes, StreamNet is purely a runtime/compiler solution. The codegen of StreamNet yields the streaming buffer used when transferring data at the runtime without changing the original model structure. As a result, StreamNet has the same output and accuracy compared to the original model.
>
> D.2 How might one redesign MCU class models to maximize the benefits of StreamNet?
>
> It is possible to use the Neural Architecture Search (NAS) method to discover new parameters of DNN models to meet the given memory budget on MCUs. Unlike the NAS method, the StreamNet can be used for existing DNN models without having to change the structure of the original model. The StreamNet also eliminates the model training processing used by the NAS. Furthermore, the StreamNet compiler also automatically analyzes the structure of the DNN model to create streaming buffers and reduces the amount of unnecessary recomputations shown on the patch-based inference. As a result, the StreamNet decreases the additional burden to manually craft a new DNN model to meet the requirement of the MCUs.
>
> D.3 Will StreamNet be open-sourced?
>
> StreamNet will be open-sourced after the paper is published.
>
> D.4 How long does it take to run the auto-tuning framework? How does it compare to black-box search methods?
>
> The StreamNet auto-tuning method takes about one minute to discover the best parameters used to reorganize the structure of the streaming buffer and fit a DNN model in the MCU. The black-box (brute-force) method firstly enumerates all combinations of the n_patch and the split_idx parameters. Second, the black-box method compiles combinations of the n_patch and the split_idx parameters with all feasible configurations of streaming levels (x or y streaming). Finally, the black-box method works out parameters that will consume the minimum amount of MACs while meeting the SRAM memory budget of the MCU. Unlike the black-box method, StreamNet leverages the Pareto Front optimization method to reduce the size of the search space. As a result, the StreamNet auto-tuning achieves a geometric mean of 1.29X speedup over the black-box method on the Apple Silicon M1 CPU by using single thread.
>
> | Benchmark            | StreamNet Auto tuning  | Brute Force  | Speedup  |
> |----------------------|------------------------|--------------|----------|
> | mcunet-vww0 (MV0)    | 51.17s                 | 83.00s       | 1.62X    |
> | mcunet-vww1 (MV1)    | 46.47s                 | 77.86s       | 1.68X    |
> | mcunet-vww2 (MV2)    | 95.42s                 | 107.44s      | 1.13X    |
> | mcunet-in0 (MI0)     | 91.68s                 | 134.30s      | 1.46X    |
> | mcunet-in1 (MI1)     | 59.17s                 | 93.85s       | 1.59X    |
> | mcunet-in2 (MI2)     | 90.87s                 | 92.39s       | 1.02X    |
> | mcunet-in3 (MI3)     | 61.11s                 | 64.95s       | 1.06X    |
> | mcunet-in4 (MI4)     | 124.40s                | 126.61s      | 1.02X    |
> | mbv2-w0.35 (MB2)     | 73.28s                 | 101.98s      | 1.39X    |
> | proxyless-w0.3 (PL)  | 65.90s                 | 79.89s       | 1.21X    |

---

> > ### Comment · Reviewer_TjRD · 2023-08-10
> > **Thank you for the response**
> >
> > The authors' response resolves my main concerns with the paper.
> >
> > I will say that the speed-up over the brute force method isn't particularly large, and the brute force method seems easier to parallelize over multiple threads or cores. Maybe worth de-emphasizing the auto tuning framework as a contribution given these results?

---

> > > ### Author Response · Authors · 2023-08-10
> > >
> > > It is good to know our responses resolve your main concerns in our paper. We will fine tune the contributions of our paper based on your comments. Thank you.

---

### Official Review · Reviewer_xxe7 · 2023-07-03

**Soundness:** 2 fair
**Presentation:** 3 good
**Contribution:** 2 fair
**Rating:** 6
**Confidence:** 4

**Summary:**

This paper introduces StreamNet, a novel approach designed to eliminate the performance bottleneck associated with patch-based inference, which incurs additional computational overheads due to overlapping patches. StreamNet comprises two techniques, StreamNet-1D and StreamNet-2D, each offering a different trade-off between memory overhead for buffering intermediate results and computational overhead reduction. The authors also propose an auto-tuning framework to derive an inference schedule, which is a composition of the two techniques, given the memory constraints of a Microcontroller Unit (MCU). Evaluation results indicate that the proposed approach significantly improves upon prior work in terms of memory usage and latency.

**Strengths:**

- Clear and comprehensive illustration of the method: The paper provides good illustrations of the proposed methods, making them easy to understand. The examples used to explain the methods are also helpful in allowing readers to understand the advantages of the proposed methods and the trade-off between design elements.
- Intuitive and straightforward methods: The proposed StreamNet-1D and StreamNet-2D techniques are intuitive and straightforward, offering a clear path to reducing computational overheads by buffering tensors shared between patches.
- Practicality and on-device evaluation: The paper demonstrates the practicality of the proposed methods through on-device evaluation, which is a strength. However, more comprehensive testing is needed.

**Weaknesses:**

- Limited experimental settings: The experimental setting does not seem to cover the common user scenario of employing patch-based inference. This limits the generalizability of the results and makes it difficult to assess the full potential of the proposed methods. Future work should include more diverse testing scenarios to fully evaluate the effectiveness and applicability of StreamNet. In my opinion, two control variables, input resolution and number of patches, are missing in the experiments.
    * Patch-based inference is favorable in settings with large input resolution where the overhead of re-computation can be amortized by the large patch size. For instance, the re-computation overhead reported in MCUNetV2 is only 10% for MobileNetV2, but such overhead appears significantly higher in this paper. What causes the discrepancy requires further discussion.
    * The number of patches used also significantly influences the re-computation overheads and memory usage. It would be intriguing to observe the trade-off between latency and memory usage achieved by the proposed method in comparison to the baselines.

**Questions:**

1. Could you provide more insight into how the current experimental setup mirrors different user scenarios, particularly those that commonly employ patch-based inference?
2. Could you provide the original latency and memory usage of the models without patch-based inference as reference data points?
3. How would varying the input resolution and the number of patches impacts the performance of StreamNet?

**Limitations:**

Yes.

---

> ### Author Rebuttal · Authors · 2023-08-09
>
> Thank you all for the valuable comments!  In our revised version, we  addressed all of the reviewer comments. The details of our reply and the changes are presented in the following.
>
> C.1 The re-computation overhead reported in MCUNetV2 is only 10% for MobileNetV2, but such overhead appears significantly higher in this paper. What causes the discrepancy requires further discussion.
>
> The MCUNet-v2 model zoo does not include the MobileNet-v2 model used in the MCUNet-v2 paper. Thus, we cannot directly compare the result reported in the MCUNet-v2 paper with that in the StreamNet paper. In addition, the re-computation overhead of the patch-based inference (MCUNet-v2) becomes worse when decreasing the peak usage of SRAM memory. The MCUNet-v2 uses the patch-based inference to increase 10% of the MACs while decreasing 87.5% of the peak SRAM memory usage compared to the layer-wise tensor memory allocation. Although the MCUNet-v2 can further decrease the SRAM memory usage of TinyML models on MCUs.  MCUNet-v2 only presents the SRAM memory usage of TinyML models when increasing 10% of the MACs.  This is because the amount of recomputation in the MCUNet-v2 will significantly increase when the SRAM memory usage of TinyML models is minimized. Unlike MCUNet-v2, the StreamNet minimizes the usage of the SRAM memory without significantly increasing the amount of MACs on TinyML models.
>
> C.2 Could you provide more insight into how the current experimental setup mirrors different user scenarios, particularly those that commonly employ patch-based inference?
>
> The usage of the SRAM memory can drop significantly in the patch-based inference by only increasing a few MACs. For instance, the original PL model contains 38.3 M of the MACs and uses 259 KB SRAM memory space. Then, the patch-based inference increases 8.8% of the MACs (41 M) and decreases the usage of the SRAM memory on the PL model to 158 KB. Unlike MCUNet-v2, the amount of MACs in the StreamNet is 39.1 M when using 92 KB SRAM memory space. As a result, the StreamNet needs fewer MACs and SRAM memory space than the patch-based inference, since the StreamNet removes the recomputation overhead of the patch-based inference while using small SRAM memory space.
>
> C.3 Could you provide the original latency and memory usage of the models without patch-based inference as reference data points?
>
> The original latency was obtained from the MCUNet-v1 that does not use the patch-based inference. To achieve a fair comparison of the latency on TinyML models, the StreamNet and the MCUNet-v1 use the same back-end library. The MCUNet-v1 reduces the peak SRAM memory usage on TinyML models by performing the in-place depthwise operation instead of using the patch-based method. Table 2 presents the StreamNet-2D gains about 25% runtime overhead when processing streaming buffers over the MCUNet-v1. Moreover, the StreamNet-2D saves about 42.3% SRAM memory usage over the MCUNet-v1 by using the patch-based inference with 2D streaming buffers.
>
> |        | Original Latency        |               | StreamNet-2D            |               |
> |--------|-------------------------|---------------|-------------------------|---------------|
> | Model  | SRAM Memory Usage (KB)  | latency (ms)  | SRAM Memory Usage (KB)  | latency (ms)  |
> | MB2    | 295                     | 367           | 66                      | 417           |
> | PL     | 259                     | 542           | 95                      | 676           |
> | MI0    | 49                      | 82            | 30                      | 99            |
> | MI1    | 96                      | 169           | 47                      | 188           |
> | MI2    | 215                     | 869           | 169                     | 1,168         |
> | MI3    | 260                     | 1,048         | 208                     | 1,444         |
> | MI4    | 416                     | 1,371         | 236                     | 1762          |
> | MV0    | 59                      | 86            | 29                      | 101           |
> | MV1    | 92                      | 159           | 44                      | 225           |
> | MV2    | 174                     | 718           | 143                     | 961           |
>
> C.4 How would varying the input resolution and the number of patches impact the performance of StreamNet?
>
> The following two figures present the latency and memory variations of the StreamNet-2D when changing the number of patches in the PL model. As illustrated in Figure I, the execution time of the PL model does not significantly fluctuate when varying the number of patches on the STM32F767 MCU. In Figure II, the SRAM memory usage is sensitive to the value of the split_idx. For instance, in Figure I, the SRAM memory usage on the setting of the (2, 12) is 50.5% smaller than the (2, 2). In addition, the SRAM memory usage also drops when increasing the number of patches. For example, in Figure II, the SRAM memory usage of the (44, 4) is 14.4% smaller than the (2, 4). Furthermore, the size of the patches will increase with the growth of the input resolution without changing the value of the n_patch and the split_idx. Hence, the StreamNet-2D requires more stream buffer space to store tensor data and the recomputation overhead of the StreamNet-2D does not increase with the growth of the input resolution.

---

> > ### Comment · Reviewer_xxe7 · 2023-08-16
> >
> > Thank you for the detailed response, which clarifies my main concerns. I am comfortable raising my rating to a weak accept.

---

### Official Review · Reviewer_XmQf · 2023-07-05

**Soundness:** 4 excellent
**Presentation:** 4 excellent
**Contribution:** 3 good
**Rating:** 6
**Confidence:** 4

**Summary:**

The patch-based inference is widely employed for TinyML models on resource-constrained microcontroller units (MCUs), which significantly reduces memory requirements compared to layer-based inference. However, path-based inference can lead to a substantial increase in Multiply-Accumulates (MACs), as it introduces a great deal of redundant computation among adjacent patches. This paper introduces StreamNet, a solution that curtails repeated computation by memorization. Furthermore, StreamNet can auto-tune the area to be skipped during path-based inference, enhancing overall efficiency.

**Strengths:**

1. The paper provides a comprehensive explanation of the benefits and drawbacks of patch-based inference.
2. The proposed method is clearly articulated and easy to understand.
3. StreamNet not only reduces MACs by leveraging the memory of the overlapped computation area, but it also introduces an auto-tuning framework to streamline these areas.
4. StreamNet outperforms the state-of-the-art methods at a relatively minor expense of additional memory space.
5. The auto-tuning framework possesses the ability to further accelerate inference automatically if more memory space is made available.

**Weaknesses:**

1. It would be better to list the potential overlapped area percentage for several commonly used tinyML models.


**Questions:**

I enjoyed reading the paper, and the idea is well present, and the gains are intuitively making sense. My only question is that what is the reuse distance of patches? Is there any case that the stream buffer is not sufficient to capture all the reuses?

---

> ### Author Rebuttal · Authors · 2023-08-09
>
> Thank you all for the valuable comments!  In our revised version, we  addressed all of the reviewer comments. The details of our reply and the changes are presented in the following.
>
> B.1 What is the reuse distance of patches?
>
> The reuse distance of the StreamNet means the distance of the data in a patch reused by another patch. For example, in Figure 3 of the StreamNet paper, the data of the patch 0 will be reused by the patch 1 in the StreamNet-1D. Thus, the reuse distance of the StreamNet-1D is 1. In addition, as illustrated in Figure 4, the data of patch 1 will be reused by patch 3 in the horizontal direction of the StreamNet-2D. Therefore, the reuse distance of StreamNet-2D in the horizontal direction is 2 because the StreamNet-2D will reuse the  data after passing two patches. That means the reuse distance in the horizontal direction of the StreamNet-2D is the value of the n_patch.
>
> B.2 Is there any case that the stream buffer is not sufficient to capture all the reuses?
>
> The StreamNet-2D typically captures all the reuses of the patch-based inference in the 2D convolution and the depthwise convolution. However, the StreamNet-2D needs more SRAM memory space to store its 2D stream buffer than the StreamNet-1D. Therefore, the StreamNet-2D will increase the peak usage of the SRAM memory and may run out of the SRAM memory space on MCUs. To address this challenge, our StreamNet mixes the StreamNet-1D and the StreamNet-2D to reduce the peak usage of the SRAM memory on MCUs. Hence, the StreamNet does not capture all the reuses when using both the StreamNet-2D and the StreamNet-1D in a Convolutional Neural Network (CNN) model.

---

> > ### Comment · Reviewer_XmQf · 2023-08-16
> >
> > The rebuttal has addressed my concerns. Thanks.

---

### Official Review · Reviewer_bp8j · 2023-07-06

**Soundness:** 2 fair
**Presentation:** 4 excellent
**Contribution:** 3 good
**Rating:** 6
**Confidence:** 5

**Summary:**

The processing of patch-based inference for MCUs induce a large number of redundant MACs against the layer-wise processing because of the overlapped processing. In order to address this problem,  this work designs StreamNet that employs the stream buffer to eliminate the redundant computation of patch-based inference. StreamNet uses 1D and 2D streaming processing and an auto-tuning framework to significantly improve the performance of patch-based inference with minimal requirements on the MCU’s SRAM memory space.

**Strengths:**

1) The proposed streamnet removes the computing redundancy in prior patch-based DNN processing framework and improves DNN inference performance significantly.

2) The paper is well organized and easy to follow. The experiments are sufficient and to the point.

**Weaknesses:**

StreamNet essentially introduces additional buffers to explore the redundant computing results induced by patched DNN processing without compromising the memory requirements. The major contribution will be the DNN computing system or implementation optimization and the novelty is relatively limited.

**Questions:**

1) According to the experiments, we notice that the performance speedup is sensitive to the different neural network architectures say kernel sizes and feature map sizes. While the benchmarks are mostly obtained from NAS and it is a bit difficult to evaluate the representation of these models. Could you provide more details about the models such as the number of layers, model sizes, and accuracy?

2) Transformer models are increasingly utilized, will the proposed framework be applicable to transformer models?

**Limitations:**

yes

---

> ### Author Rebuttal · Authors · 2023-08-09
>
> Thank you all for the valuable comments!  In our revised version, we  addressed all of the reviewer comments. The details of our reply and the changes are presented in the following.
>
> A.1 Could you provide more details about the models such as the number of layers, model sizes, and accuracy?
>
> Our benchmark models were obtained from the MCUNet model zoo and were used to perform the real-world applications such as visual wake word (VWW) and image classification on MCUs. The MCUNet uses the Neural Architecture Search (NAS) to fine tune the accuracy, memory usage, and the amount of MACs of TinyML models to meet the requirements of MCUs. In the following table, each TinyML model comprises multiple layers. The model size and the top-1 accuracy of the MCUNet models are presented in Table 1. These MCUNet models can serve as representatives to real-world TinyML applications on MCUs, which faithfully reflect the performance improvements of our StreamNet.
>
> | Benchmark            | number of layers  | model sizes (MB)  | Top-1 Accuracy  |
> |----------------------|-------------------|-------------------|-----------------|
> | mcunet-vww0 (MV0)    | 55                | 0.37              | 87.3%           |
> | mcunet-vww1 (MV1)    | 51                | 0.43              | 88.9%           |
> | mcunet-vww2 (MV2)    | 83                | 0.64              | 91.8%           |
> | mcunet-in0 (MI0)     | 51                | 0.75              | 40.4%           |
> | mcunet-in1 (MI1)     | 55                | 0.64              | 49.9%           |
> | mcunet-in2 (MI2)     | 67                | 0.73              | 60.3%           |
> | mcunet-in3 (MI3)     | 67                | 0.74              | 61.8%           |
> | mcunet-in4 (MI4)     | 63                | 1.73              | 68.0%           |
> | mbv2-w0.35 (MB2)     | 64                | 0.75              | 49.0%           |
> | proxyless-w0.3 (PL)  | 76                | 0.75              | 56.2%           |
>
> A.2 Will the proposed framework be applicable to transformer models?
>
> Since the receptive field in the convolution operation works in the sliding window manner, the patch-based inference can divide the entire input into multiple small tiles and reduces the peak SRAM memory usage by only storing one of the small patches. Unlike the convolution operation, in the transformer model, the receptive field covers the entire input. Thus, the tiling method used by the patch-based inference does not work in transformer models. StreamNet aims to decrease the amount of the recomputation caused by the overlapping patches shown on the patch-based inference, which is not applicable to transformer models. Hence, we need to redesign our StreamNet to decrease the SRAM memory usage on transformer models.

---

### Author Rebuttal · Authors · 2023-08-09

Thank you all for the valuable comments!  In our revised version, we  addressed all of the reviewer comments. The details of our reply and the changes are presented in the following.

C.1 The re-computation overhead reported in MCUNetV2 is only 10% for MobileNetV2, but such overhead appears significantly higher in this paper. What causes the discrepancy requires further discussion.

The MCUNet-v2 model zoo does not include the MobileNet-v2 model used in the MCUNet-v2 paper. Thus, we cannot directly compare the result reported in the MCUNet-v2 paper with that in the StreamNet paper. In addition, the re-computation overhead of the patch-based inference (MCUNet-v2) becomes worse when decreasing the peak usage of SRAM memory. The MCUNet-v2 uses the patch-based inference to increase 10% of the MACs while decreasing 87.5% of the peak SRAM memory usage compared to the layer-wise tensor memory allocation. Although the MCUNet-v2 can further decrease the SRAM memory usage of TinyML models on MCUs.  MCUNet-v2 only presents the SRAM memory usage of TinyML models when increasing 10% of the MACs.  This is because the amount of recomputation in the MCUNet-v2 will significantly increase when the SRAM memory usage of TinyML models is minimized. Unlike MCUNet-v2, the StreamNet minimizes the usage of the SRAM memory without significantly increasing the amount of MACs on TinyML models.

C.2 Could you provide more insight into how the current experimental setup mirrors different user scenarios, particularly those that commonly employ patch-based inference?

The usage of the SRAM memory can drop significantly in the patch-based inference by only increasing a few MACs. For instance, the original PL model contains 38.3 M of the MACs and uses 259 KB SRAM memory space. Then, the patch-based inference increases 8.8% of the MACs (41 M) and decreases the usage of the SRAM memory on the PL model to 158 KB. Unlike MCUNet-v2, the amount of MACs in the StreamNet is 39.1 M when using 92 KB SRAM memory space. As a result, the StreamNet needs fewer MACs and SRAM memory space than the patch-based inference, since the StreamNet removes the recomputation overhead of the patch-based inference while using small SRAM memory space.

Unlike the MCU used in the StreamNet paper, there is another user scenario when using the STM32L412 MCU that contains 256 KB SRAM memory and 1MB flash memory. The peak SRAM memory usage of the Ml4 model is 426 KB, and the amount of MACs is 126 M when using the Google TensorFlow Lite for Microcontroller (TFLM) with the layer-wise tensor memory allocation. The TFLM will cause the out-of-memory exception on the STM32L412 MCU, since the SRAM memory usage of the Ml4 model is larger than the SRAM memory space on the STM32L412 MCU. The MCUNet-v2 performs the patch-based inference to reduce the use of the SRAM memory on the MCU. Thus, in the MCUNet-v2, the Ml4 model only uses 129 KB SRAM memory, but the amount of MACs becomes 1942 M when using the patch-based inference. To fit the SRAM memory constraint, in the StreamNet-2D, the Ml4 model uses 241 KB SRAM memory space with 126M MACs. The StreamNet-2D reduces 15.41X MACs by capturing all reuses in the Ml4 model. In addition, in the StreamNet-1D, the amount of MACs is 354 M, and the peak SRAM memory usage is 150 KB. The outcome of the StreamNet-1D is similar to the result of the MCUNet-v2 in the Ml4 model. Typically, we can adjust parameters of the MCUnet-v2 to increase the usage of the SRAM memory and decrease the amount of the recomputation on overlapping patches. For instance, to meet the size of the SRAM memory on the STM32L412 MCU, we choose that value of the (5, 19) in the (n_patch, split_idx) parameter that yields the usage of the SRAM memory closest to the memory constraint on the STM32L412 MCU. Thus, in such MCUNet-v2 parameters, the amount of MACs in the Ml4 model becomes 458 M, and the peak SRAM memory usage is 239 KB. Therefore, it is possible to adjust values of the patch-based inference parameters to meet the memory constraint of the MCUs. However, the patch-based inference exposes the extremely high recomputation overhead when lowering the peak SRAM memory usage or using the MCU with a small SRAM memory. Unlike the patch-based inference, the StreamNet eliminates the large portion of the recomputation overhead shown in the patch-based inference without significantly increasing the usage of the SRAM memory.

C.4 How would varying the input resolution and the number of patches impact the performance of StreamNet?

The following two figures present the latency and memory variations of the StreamNet-2D when changing the number of patches in the PL model. As illustrated in Figure I, the execution time of the PL model does not significantly fluctuate when varying the number of patches on the STM32F767 MCU. In Figure II, the SRAM memory usage is sensitive to the value of the split_idx. For instance, in Figure I, the SRAM memory usage on the setting of the (2, 12) is 50.5% smaller than the (2, 2). In addition, the SRAM memory usage also drops when increasing the number of patches. For example, in Figure II, the SRAM memory usage of the (44, 4) is 14.4% smaller than the (2, 4). Furthermore, the size of the patches will increase with the growth of the input resolution without changing the value of the n_patch and the split_idx. Hence, the StreamNet-2D requires more stream buffer space to store tensor data and the recomputation overhead of the StreamNet-2D does not increase with the growth of the input resolution.

---

### Decision · Program_Chairs · 2023-09-21

**Decision:**

Accept (poster)

**Comment:**

The submitted paper presents StreamNet, a framework designed to optimize patch-based DNN inference on microcontroller units (MCUs) by reducing computational redundancy. StreamNet achieves this using stream buffers and offers an auto-tuning framework for further optimization. The paper aims to balance memory and latency, providing a significant improvement over previous methods. The paper has garnered four "Weak Accept" ratings from the reviewers, with scores generally indicating good presentation and contribution.

Given the above considerations and the "Weak Accept" consensus among the reviewers, the paper is a candidate for acceptance. The reviewers agree on the paper's strengths in presentation and practical contributions. It is advised that the authors take into consideration the feedback on the experimental setup and general applicability for future revisions or presentations. All reviewers have moderate to high confidence in their assessments, adding weight to the decision.